# The Impact of Deep Core Muscle System Training Through Virtual Reality on Selected Posturographic Parameters

**DOI:** 10.3390/jfmk10020185

**Published:** 2025-05-21

**Authors:** Jakub Čuj, Denisa Lenková, Miloslav Gajdoš, Eva Lukáčová, Michal Macej, Katarína Hnátová, Pavol Nechvátal, Lucia Demjanovič Kendrová

**Affiliations:** Department of Physiotherapy, Faculty of Health Care, University of Prešov in Prešov, 080 01 Prešov, Slovakia; denisa.lenkova@smail.unipo.sk (D.L.); miloslav.gajdos@unipo.sk (M.G.); eva.lukacova@unipo.sk (E.L.); michal.macej@unipo.sk (M.M.); katarina.hnatova@unipo.sk (K.H.); pavol.nechvatal@unipo.sk (P.N.); lucia.kendrova@unipo.sk (L.D.K.)

**Keywords:** virtual reality, Icaros^®^, deep muscle system, posturography

## Abstract

**Objective:** The aim of this study was to investigate the immediate effects of deep core muscle training in the plank position, using the Icaros^®^ system, integrated with virtual reality (VR), on selected posturographic parameters. **Methods:** To meet the stated objective, we utilized the Icaros^®^ therapeutic system (Icaros GmbH, Martinsried, Germany) for VR-based exercise. The posturographic parameters were measured using the FootScan^®^ force platform (Materialise Motion, Paal, Belgium). A representative sample of 30 healthy participants, 13 females and 17 males (age: 22.5 ± 2.1 years; weight: 65 ± 2.9 kg; height: 1.68 ± 0.4 m; BMI: 23.04 ± 1.75) was included in the study. All participants had no prior experience with VR. The selected posturographic parameters were the ellipse area (mm^2^) and traveled distance (mm), assessed four times at five-minute intervals, following a 15 min VR-based training session on the Icaros^®^ system. **Results:** The results revealed that the participants experienced a sense of instability after completing the 15 min VR session, as objectively demonstrated by changes in the measured parameters. Both the ellipse area and traveled distance showed a worsening trend during the first three measurements: immediately post-exercise, at 5 min, and at 10 min post-exercise. A downward trend was observed in the fourth measurement, taken 15 min after exercise. Statistically significant differences were found between both parameters: ellipse area (*p* = 0.000) and traveled distance (*p* = 0.000). Post hoc analysis further confirmed significant differences between the time points. **Conclusions:** Based on the findings, it is recommended that trainers and physiotherapists supervising athletes or patients using the Icaros^®^ VR system allow for a minimum rest period of 15 min in a seated or lying position following exercise. This recovery period appears essential to mitigate the sensation of instability and to reduce the risk of complications or injury due to potential falls.

## 1. Introduction

Postural stability is essential for maintaining balance and proper body biomechanics. It is influenced by several factors, including the function of the deep core muscle system, sensory integration, and motor control. In recent years, there has been growing interest in innovative approaches to sports training, rehabilitation, and physiotherapy, with virtual reality emerging as one such method. This technology offers new opportunities for enhancing muscle activity and delivering therapeutic interventions through immersive therapy, which can effectively stimulate the deep core musculature. The deep core muscle system plays a crucial role in spinal stabilization and maintaining correct posture [1]. Dysfunction in this system may result in impaired balance, musculoskeletal pain, and an increased risk of falls [1,2,3]. Traditional methods of strengthening deep muscles include stabilization exercises, training on unstable surfaces, and proprioceptive training [4,5,6]. Novel technological solutions, such as VR, allow for a more interactive and potentially more effective approach to core training.

The interactive and immersive nature of virtual reality can enhance patient motivation and improve adherence to training or rehabilitation protocols [7]. VR technology enables users to experience presence within a computer-generated three-dimensional virtual environment so realistically that they may perceive themselves to be physically part of it [8,9]. Simulated environments created through VR can provide varying degrees of perceived realism and presence. The more realistic the virtual environment appears to the user, the more intense the immersive experience becomes. This experience depends on the number of physiological senses stimulated, the level of interaction with the virtual environment, and the degree of isolation from external stimuli, such as light or ambient noise [10].

In physiotherapy, VR has also been proven to be an effective tool for managing acute pain [11]. In the treatment of chronic patients, it can be utilized for cognitive rehabilitation, by applying principles of cognitive–behavioral therapy [12]. Furthermore, the positive effects of VR-based kinesiotherapy have been demonstrated in the re-education of balance skills and gait patterns in individuals diagnosed with multiple sclerosis, Parkinson’s disease, stroke, and spinal cord injuries [13,14,15,16].

Several studies have confirmed that exposure to virtual reality (VR) may induce VR sickness, characterized by symptoms such as nausea, disorientation, a sense of instability, and impaired balance [17]. According to current evidence, no studies have specifically investigated the effects of plank-position exercises in VR on the function of the deep core muscle system, using the Icaros^®^ device.

Since no prior study has been published on the effects of exercising using the Icaros^®^ system, in combination with virtual reality (VR), and based on the existing findings related to VR-induced symptoms, such as motion sickness, we decided to conduct an initial investigation into the effects of this type of exercise in VR on individuals without any diagnosed medical conditions. This approach will allow us to assess how this combined therapy affects the human body and motor control immediately after exercise in healthy individuals. The results can then be used to adapt and tailor the exercise protocols for patients with specific diagnoses, as well as to identify which conditions are suitable for this form of exercise.

The aim of this study is to analyze the impact of deep core muscle activation during virtual reality exercise on selected posturographic parameters in healthy individuals immediately after training. The main motivation for conducting this study was the recurrent reports of instability and dizziness observed in previous clients following their first VR training session on the Icaros^®^ system. This research explores the potential and applicability of VR in clinical practice, with findings that may contribute to a better understanding of the influence of VR on postural control.

## 2. Materials and Methods

The present research is designed as an experimental study aimed at evaluating the immediate effect of exercise using the Icaros^®^ system in a virtual reality environment on postural stability, thereby examining the influence of VR on postural control mechanisms.

### 2.1. Participants

The study sample consisted of 30 healthy participants, namely 17 males and 13 females, with the following characteristics: age 22.5 ± 2.1 years; weight 65 ± 2.9 kg; height 1.68 ± 0.04 m; and BMI 23.04 ± 1.75. All the participants were free from any medical conditions or coordination disorders that could have influenced the study outcomes. The selected participants were recreational athletes with no prior experience of VR-based exercise. Each participant received comprehensive information regarding the research objectives and measurement procedures. Prior to participation, all subjects voluntarily signed a written informed consent form. They were informed of their right to withdraw from the study at any time and assured that the data would be processed and published with full anonymity. The study was approved by the institutional ethics committee under approval number 02/2024. The research protocol was conducted in accordance with the principles of the Declaration of Helsinki and other applicable data protection regulations.

### 2.2. Measuring Protocol

To examine the effects of deep core muscle training using virtual reality, the intervention was conducted using the Icaros^®^ virtual reality training and therapeutic system (Icaros GmbH, Martinsried, Germany). The posturographic parameters were assessed using the FootScan^®^ pressure platform (Materialise Motion, Paal, Belgium). Both systems are non-invasive and pose no health risks to the participants. The research procedure consisted of two main components. The therapeutic phase involved participants activating their deep core muscle system to control the Icaros^®^ device in a VR environment. Immediately following this, the diagnostic phase was conducted, in which the posturographic parameters were measured and evaluated using the pressure platform. This allowed for the assessment of the immediate impact of the VR-based intervention on the participant’s postural control.

#### 2.2.1. Icaros^®^

Icaros^®^ is a modern and unique therapeutic device that combines core strengthening with virtual reality training (Figure 1). It is specifically designed to facilitate assisted, comprehensive training of the abdominal and back core zones, offering a wide range of training modalities that ensure both safety and high training efficacy. The system enables activation and simultaneous training of the core musculature, along with other muscle groups, using the so-called plank position, which is widely recognized as one of the most effective and essential positions for engaging the abdominal and spinal cord regions. The device allows therapists or trainers to create individualized programs, tailored to each athlete or client, enabling the repeated application of targeted exercises, based on specific issues or desired training goals. VR has a broad spectrum of applications in regard to enhancing the deep core stability (DCS) function in both athletes and the general population, as well as in pain management for the lumbar spine [8,9]. Moreover, it has shown potential in regard to improving cognitive function in patients with systemic neurological conditions, such as multiple sclerosis [18], Parkinson’s disease [14], and others.

By adjusting the range of motion and reactivity levels of the device to different virtual scenarios, various levels of exercise difficulty can be implemented, thereby simulating an unstable environment that challenges multiple sensory modalities. The Icaros^®^ system is characterized by a high safety standard, making it suitable, even relatively demanding training sessions, for individuals without prior experience of this type of exercise. This unique combination allows for the execution of advanced exercises by users, regardless of their previous training background.

#### 2.2.2. FootScan^®^

The FootScan^®^ pressure platform has been widely used in numerous studies for the objective assessment of foot function, gait analysis, and postural control. It provides quantitative data on plantar pressure distribution [19,20], posturographic parameters, and the projection of the centre of gravity onto the base of support [21], as well as gait cycle diagnostics [22,23].

In this study, we focused on the parameters: ellipse area and traveled distance. The ellipse area (mm^2^) is defined as the area encompassing 95% of the Centre of Pressure (CoP) trajectory during foot–ground contact. This parameter is commonly used to evaluate postural stability and balance during standing or walking. A larger ellipse area typically indicates postural instability or impaired movement control, whereas a smaller area reflects better stability. The traveled distance (mm) represents the total length of the CoP trajectory recorded during static or dynamic standing and gait assessments. Higher values may indicate inefficient postural control or compensatory movements to maintain balance. It is important to note that there are no universal normative values for these parameters, as they vary based on factors such as age, sex, physical fitness, and health status (Figure 2).

### 2.3. The Measurement Procedure

Data collection was conducted in the indoor facilities at the Faculty of Health Sciences, within the kinesiology laboratory of the Department of Physiotherapy, where both the therapeutic VR system, Icaros^®^, and the diagnostic FootScan^®^ system are located. The controlled indoor environment provided uniform testing conditions for all the participants. Due to the time-consuming nature of the measurements, data collection was carried out progressively throughout November 2024. The measurement procedure was as follows: upon arrival at the laboratory, each participant changed into comfortable sportswear. The participants received instructions on how to operate the Icaros^®^ system, followed by individual calibration of the equipment, including a practical trial to determine the range of motion in all directions. The optimal position was established, with support under the forearms and lower legs, while the core remained unsupported. Virtual reality goggles were individually adjusted for clarity and comfort. After mastering the control of the Icaros^®^ system, participants took a short break to rest and prepare for the 15 min training session. During this break, the participants were introduced to the scenario and virtual environment they would be navigating. The same standardized VR scenario was used for all the participants. It involved a simulated flight experience in a mountainous setting (Dolomites), where the participant, controlling an avatar in a wingsuit, was required to fly through checkpoints, each associated with a score, thus serving as a motivational element to perform well. The VR flight session on the Icaros^®^ system lasted 15 uninterrupted minutes. For added safety, the system was connected to an external monitor, allowing researchers to observe the participant’s VR experience and monitor the session in real time (Figure 2). Immediately after completing the 15 min exercise, the goggles were removed, and the participant stepped directly onto the FootScan^®^ pressure plate. The posturographic measurement began immediately and lasted for 30 s. During this time, the participant stood motionless in the centre of the platform, arms relaxed alongside their body, their feet a shoulder-width apart, and their eyes open (Figure 3). This posturographic assessment was repeated three more times at 5 min intervals. Thus, each participant underwent four measurements in total: immediately after the VR training, and then after 5, 10, and 15 min after. Between each measurement, the participants were not allowed to move freely and were required to remain seated on a chair. The same protocol was followed for all the participants, and all data collection sessions proceeded without complications. Subjectively, most participants reported feelings of instability immediately after the VR session using the Icaros^®^ system. The entire data collection process concluded with the fourth posturographic measurement, after which the participants changed their clothes and exited the laboratory.

### 2.4. Statistical Analysis

All the numerical data were processed using Statistica 13.3 software (Hamburg, Germany). The normality of the data distribution was assessed using the Shapiro–Wilk test (Table 1). Based on the results of this normality testing, we selected the appropriate statistical methods to evaluate the significance of the differences between the measurements. For data with a normal distribution, a repeated measures ANOVA was used. In cases where normality was violated, the non-parametric Friedman test was applied as an alternative. To perform a more detailed analysis and determine statistically significant differences between individual measurements, post hoc tests were conducted. For normally distributed data, Tukey’s post hoc test was used, while the Bonferroni correction was applied for data that did not meet the assumption of normality.

## 3. Results

By evaluating the parameters, we can demonstrate the impact of deep core muscle training on the Icaros^®^ therapeutic device in virtual reality on motor stability and coordination of the body after therapy. These parameters were analyzed across the entire sample of participants and, subsequently, evaluated separately for males and females.

Table 2 presents the results of the statistical evaluation of the selected posturographic parameter, ellipse area, for all the participants. The results indicate a deterioration of this parameter during the first, second, and third measurements. In the final measurement, taken 15 min after the completion of the therapy on the Icaros^®^ VR system, a decrease in ellipse area values was observed. The results are supported by a statistically significant value (*p* ≤ 0.05), indicating a significant difference between the measurements. A graphical representation of the results is shown in Figure 4.

Table 3 presents the results of the post hoc analysis for the ellipse area parameter. This analysis compares the individual measurements and evaluates statistically significant differences. Significant differences were observed when comparing the 1st, 2nd, and 3rd measurements with the 4th measurement. No statistically significant differences were found between the remaining measurements.

Table 4 provides the results of a basic statistical evaluation of the ellipse area parameter, involving a comparison between the different genders. The results indicate that the average values of this parameter are higher in males, suggesting that the male population may require a longer period to stabilize the observed parameters and reduce the sensation of instability.

Table 5 presents the results of the statistical evaluation of the selected posturographic parameter, traveled distance, for all the participants. The values of this parameter showed a slight deterioration in the first three measurements, as demonstrated by the median values. Improvement and stabilization occurred in the fourth and final measurement, during which the participants also reported feeling subjectively more stable. The results are supported by a statistically significant value (*p* ≤ 0.05), indicating a significant difference between the measurements. A graphical representation of the results is provided in Figure 5.

Table 6 presents the results of the post hoc analysis of the traveled distance parameter. This analysis compares the individual measurements with each other and evaluates the statistically significant differences observed when comparing the 1st, 2nd, and 3rd measurements with the 4th. No statistically significant differences were observed when comparing the other measurements with each other.

Table 7 provides the results of a basic statistical evaluation of the traveled distance parameter when comparing men and women. The results suggest that the average values for this parameter are higher in men, which implies that the male population overall reacts to therapy using the Icaros^®^ system in VR with worsened coordination.

## 4. Discussion

In this study, we tested the response of a healthy human organism to the use of the Icaros^®^ system, combined with VR, immediately after exercise; a study that has not yet taken place. The statistical analysis of the obtained data confirmed that a 15 min training session on this system significantly affects the coordination and stability of a healthy person, causing one of the symptoms of VR sickness. After completing the training on Icaros^®^, the participants subjectively felt unstable, and this subjective assessment was confirmed by objective measurements taken by the pressure platform, which we performed four times, with 5 min breaks. Statistically significant differences were observed for both monitored parameters, ellipse area (*p* = 0.000) and traveled distance (*p* = 0.000) (Table 2 and Table 5). In the detailed post hoc analysis, which aimed to determine the differences between the individual measurements, statistically significant differences were confirmed only when comparing the first, second, and third measurements with the fourth (Table 3 and Table 6). Similar results were observed in studies that recorded changes in static balance after playing VR games with a fixed or non-fixed background, where significant deterioration in balance was observed [24]. In our study, the exercise was performed during a VR game with a non-fixed background, which may have contributed to the deterioration of the participant’s stability after exercise.

A variety of VR systems are utilized globally, particularly in rehabilitation, sports, and fitness environments [25]. In regard to rehabilitation, the primary focus is on neurorehabilitation for patients with neurological diagnoses (such as multiple sclerosis, Parkinson’s disease, or stroke), aimed at improving the overall quality of life and cognitive functions in elderly patients [13,14,16,18,26]. Another major area where VR therapy is significantly utilized is in pain management [8,9,10,11,27]. Athletes use VR systems due to the broad range of gaming scenarios and experiences available to enhance their performance and as injury prevention [28,29]. The advantage lies in the numerous active sports games that VR offers, which can engage individuals and, consequently, improve their physical health [25,30,31,32]. VR tools were also employed during the global COVID-19 pandemic, wherein athletes used them for training at home [33]. However, there are studies that provide various training plans [34,35] or reference systems and methodological frameworks [36] to prevent nausea, oculomotor deficits, disorientation, and VR sickness caused by using other VR systems [37]. These studies, however, yield conflicting results.

Virtual reality (VR) sickness and its symptoms are mostly explained through two main theories: the sensory conflict theory and the postural instability theory. According to sensory conflict theory, motion sickness occurs due to a conflict between visual perception and signals from the vestibular system, leading to a perceptual mismatch [38]. On the other hand, postural instability theory attributes the feeling of instability to difficulties in maintaining a stable body position [39]. Beginners in virtual reality are, therefore, more likely to experience VR sickness symptoms, due to their lesser ability to maintain a stable posture. Based on these theories, several studies have been conducted to alleviate VR-induced sickness, often through the use of software solutions. One such approach involves visual effects. Groth et al. (2021) implemented techniques such as peripheral blur and narrowing of the field of view within the content, and the experimental results confirmed their effectiveness in reducing VR sickness [40]. In a study by Nie et al. (2019), a dynamically changing blur effect was introduced in first-person perspective games, which, based on experimental data, proved to be an effective solution to reduce VR sickness [41]. Another study presents design elements aimed at alleviating cybersickness, such as movement styles, panel placement, laser use, and the background of panels [42].

The use of the Icaros^®^ system, combined with VR, has been shown to be an effective approach for training and therapy in terms of the deep muscular system. Furthermore, this training system significantly impacts the balance abilities and stability of healthy individuals immediately after the therapy. We hypothesize that the feeling of instability experienced by participants using the Icaros^®^ system is due to a combination of factors, as described by the theories on VR sickness, such as the initial VR experience, the use of VR goggles, prolonged engagement in the plank position, and the exercise involved in controlling the Icaros^®^ system. By engaging in the exercise using the Icaros^®^ system, in combination with VR, we influenced both components described in the theories of VR sickness symptoms. On one hand, the sensory component was affected by the use of VR goggles, wherein the participants were exposed to multiple sensory stimuli that could impact the function of their vestibular system. On the other hand, participants remained in the plank position throughout the exercise, which required significant activity from the deep muscular system, thus influencing the second theory related to postural instability. The prolonged stay in the plank position subjected the deep muscular system to demanding conditions, as the recommended duration for holding this position for healthy individuals aged 18–25 years typically ranges from 60 to 100 s [43,44].

An important question that remains is how real patients with specific diagnoses would respond to a similar exercise routine using this system, within a VR environment. Further research involving patients with neurological or musculoskeletal disorders, such as those with chronic low back pain, sciatica, or even those undergoing post-stroke rehabilitation, would be necessary to determine whether the therapeutic effect seen in healthy individuals translates to clinical populations. Additionally, examining the adaptation process to the plank position and VR could help in designing tailored therapies for different patient groups.

## 5. Conclusions

In conclusion, we can state that after exercising using the Icaros^®^ system, in a VR environment, changes in motor control and imbalance occur. It is essential to allow clients who are using this system in VR for the first time to sit or lie down for at least 15 min after completing the training. One of the significant advantages of VR is its ability to combine therapy or training with experiential therapy, which motivates athletes and patients to engage in repeated exercises. Another benefit is the ability to introduce the client to various gaming scenarios and virtual environments, wherein they have specific tasks or goals to achieve. Based on the study results, we recommend the following measures for clinical implementation of therapy using the Icaros system and virtual reality (VR):Gradual Adaptation to the Icaros^®^ and VR System:
For healthy people and patients using the Icaros^®^ and VR system for the first time, individual responses to the virtual environment should be considered. Adequate time for resting after exercise should be provided if necessary to prevent adverse effects, such as dizziness or a temporary loss of balance. The pace of adaptation to VR should be tailored to each individual;
2.Integration into Rehabilitation Protocols:
The Icaros^®^ system can serve as a valuable tool in rehabilitation, particularly in improving motor function and balance in patients recovering from injuries or with neurological disorders. Healthcare professionals should consider incorporating VR into rehabilitation programs to offer patients a diverse and motivating environment for enhancing their functional abilities;
3.Customization of Icaros^®^ VR Exercises to Individual Needs:
It is essential to adjust the intensity and type of VR exercises based on the patient’s condition and progress. For example, for patients with impaired balance or motor dysfunction, it may not be appropriate to initiate therapy using this system. It is recommended to start with simpler tasks and gradually increase the difficulty of the exercises;
4.Safety Considerations:
Prior to using Icaros^®^ and VR systems, it is crucial to assess the patient’s ability to move safely within the virtual environment. Clinical professionals should ensure appropriate exercise conditions, such as balanced lighting and the removal of physical obstacles, to minimize the risk of injury during therapy.

## 6. Limitations of the Study

One of the limitations of this study could be the fact that it was the participants’ first experience with the Icaros^®^ system and VR. Another limitation is the lack of reference values for the posturographic parameters being monitored, meaning we can only compare the changes in these parameters as a result of the therapy. The study also did not account for the participants’ fatigue during the measurements. However, the therapy itself was not physically strenuous, and the fatigue experienced was considered acceptable. Other limitations are the absence of a control group and the sample size. To achieve broader generalizability, groups of 100 or more participants would be necessary. The results of this study are applicable to the population with the characteristics described in the research sample section.

## Figures and Tables

**Figure 1 jfmk-10-00185-f001:**
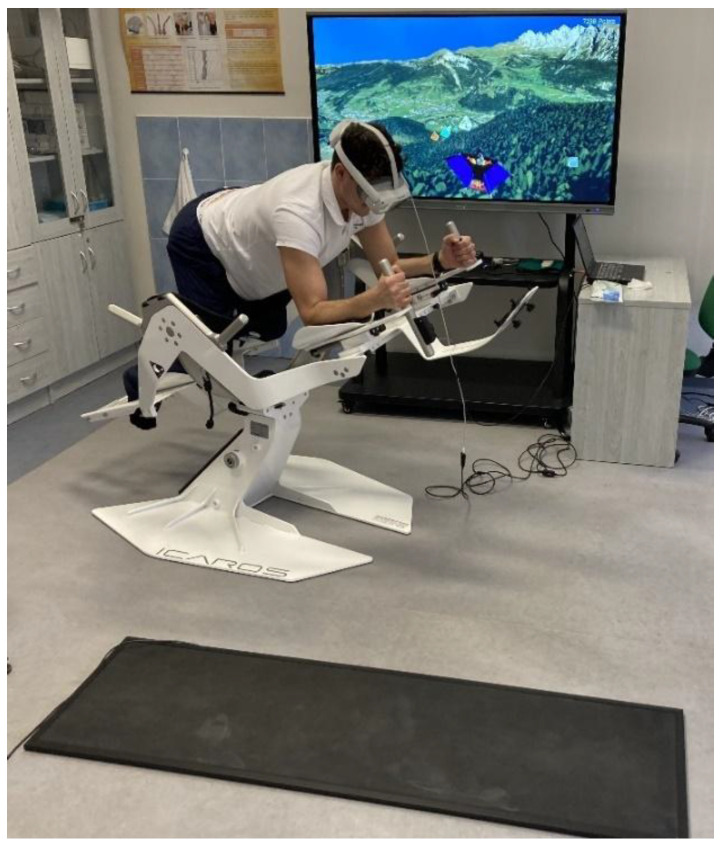
Therapy session on Icaros^®^ in a VR environment.

**Figure 2 jfmk-10-00185-f002:**
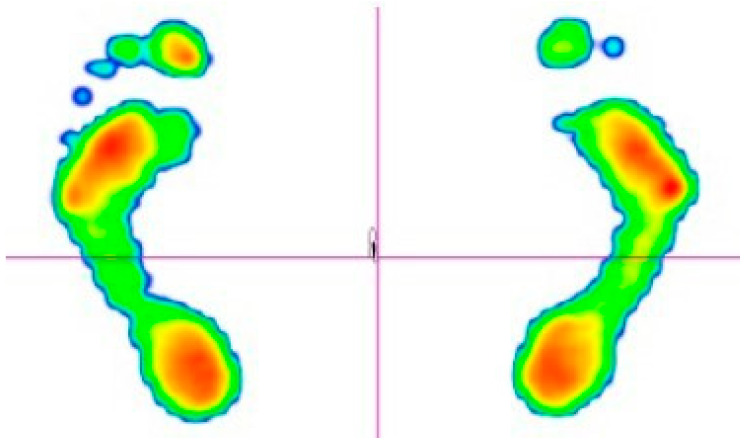
Posturographic diagnostics: ellipse area and traveled distance.

**Figure 3 jfmk-10-00185-f003:**
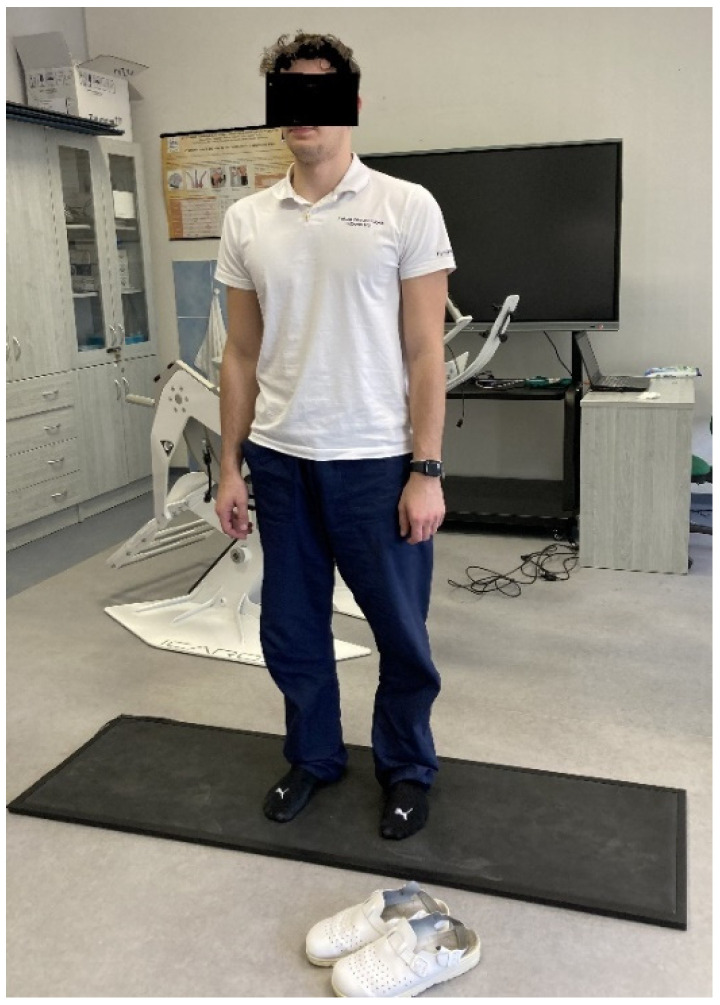
FootScan^®^ measurement.

**Figure 4 jfmk-10-00185-f004:**
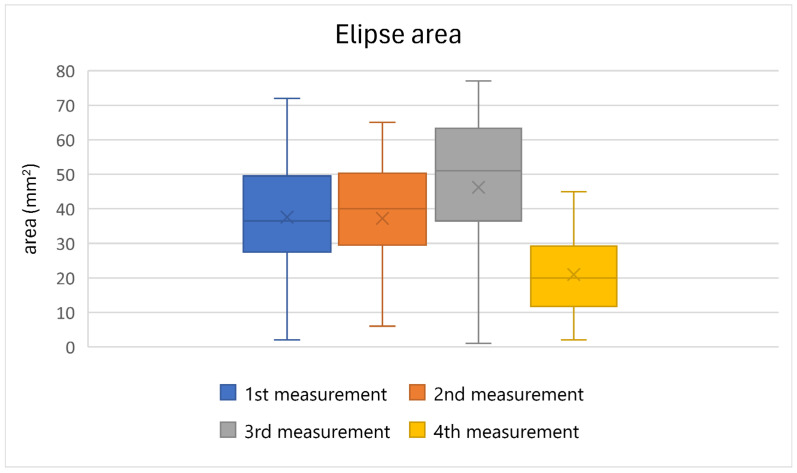
A graphical representation of the average ellipse area values of all the participants across the individual measurements.

**Figure 5 jfmk-10-00185-f005:**
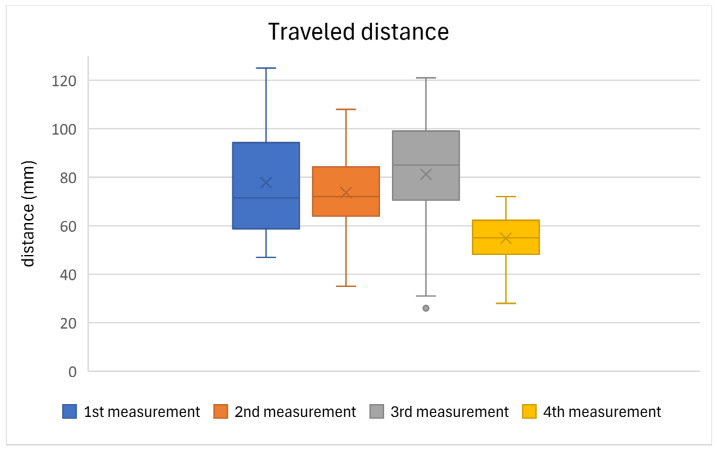
A graphical representation of the mean traveled distance values of all the participants across the individual measurements.

**Table 1 jfmk-10-00185-t001:** Data normality test results.

*n* = 30	Ellipse Area (mm^2^)		Traveled Distance (mm)	
**1st measurement**	*p* = 0.951	normality confirmed	*p* = 0.023	normality unconfirmed
**2nd measurement**	*p* = 0.119	normality confirmed	*p* = 0.964	normality confirmed
**3rd measurement**	*p* = 0.094	normality confirmed	*p* = 0.456	normality confirmed
**4th measurement**	*p* = 0.320	normality confirmed	*p* = 0.475	normality confirmed

**Table 2 jfmk-10-00185-t002:** The results of posturography evaluation (ellipse area mm^2^) of all the participants.

Ellipse Area (mm^2^); *n* = 30				
Measurements	1st	2nd	3rd	4th
**average**	37.5	37.2	46.2	20.9
**median**	36.5	40	51	20
**max**	72	65	77	45
**min**	2	6	1	2
**sd**	16.6	16.9	20.6	11.9
***p *value**	0.000			
**F value**	11.38			

**Table 3 jfmk-10-00185-t003:** The results of post hoc test for the ellipse area.

Compared Measurements	*p* Value
1st measurement vs. 4th measurement	0.001
2nd measurement vs. 4th measurement	0.002
3rd measurement vs. 4th measurement	0.001

**Table 4 jfmk-10-00185-t004:** Comparison of ellipse area (mm^2^) results between men and women.

	Women				Men			
Measurements	1st	2nd	3rd	4th	1st	2nd	3rd	4th
**average**	39.3	34.1	36.6	13.6	36.2	39.5	53.4	26.5
**median**	44	35	38	12	36	40	55	25
**max**	72	63	71	30	62	65	77	45
**min**	2	6	1	2	7	10	15	4
**sd**	19.2	19.8	22.5	7.9	14.2	13.7	15.4	11.4

**Table 5 jfmk-10-00185-t005:** The results of posturography evaluation (traveled distance) of all the participants.

Traveled Distance (mm); *n* = 30				
Measurements	1st	2nd	3rd	4th
**average**	77.8	73.6	81.2	54.8
**median**	71.5	72	85	55
**max**	125	108	142	72
**min**	47	35	26	28
**sd**	21.4	16.2	26.2	10.3
***p *value**	0.000			
**Test statistic value**	46.52			

**Table 6 jfmk-10-00185-t006:** The results of post hoc test for the traveled distance.

Compared Measurements	*p* Value
1st measurement vs. 4th measurement	0.000
2nd measurement vs. 4th measurement	0.002
3rd measurement vs. 4th measurement	0.000

**Table 7 jfmk-10-00185-t007:** A comparison of the traveled distance results between men and women.

	Women				Men			
Measurements	1st	2nd	3rd	4th	1st	2nd	3rd	4th
**average**	85.1	70.3	65.7	49.7	72.1	76.2	93.1	58.7
**median**	85	70	66	52	66	75	89	61
**max**	125	105	121	64	125	108	142	72
**min**	47	35	26	28	54	51	74	44
**sd**	23.8	17.7	28.4	10.6	17.4	14.4	16.4	8.2

## Data Availability

Data are available on request at the following email address: jakub.cuj@unipo.sk.

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
