# Peer review of "The Impact of Deep Core Muscle System Training Through Virtual Reality on Selected Posturographic Parameters"

_jfmk, 2025, doi:10.3390/jfmk10020185_

Round 1
Reviewer 1 Report
Comments and Suggestions for Authors
This article offers novel insights into the integration of virtual reality (VR) in rehabilitation, specifically targeting the activation and training of the deep core muscle system. The authors present compelling data demonstrating the potential of VR-based interventions to influence posturographic parameters, adding a valuable contribution to the growing body of evidence supporting innovative therapeutic approaches.
One of the article’s key strengths is its identification of a possible short-term complication following the application of VR in therapeutic settings—namely, a temporary decrease in balance performance. This observation is important for clinical practice, as it highlights the necessity of carefully considering the timing of VR-based exercises, particularly when they are integrated into rehabilitation protocols for individuals with impaired postural control.
Moreover, the study offers practical time-based recommendations for physiotherapists utilising VR tools, emphasising when it may be most appropriate to schedule or avoid specific activities post-intervention. This aspect significantly enhances the clinical relevance of the research and provides an evidence-based framework for optimising therapeutic outcomes.
While the article generally meets high standards in terms of structure and scientific rigour, a few areas could benefit from minor revisions:
-
Figure 1 (page 3): There is a discrepancy between the figure referenced in the text (section 2.2.1, line 104) and the actual Figure 1 presented in the article. This should be corrected to ensure clarity and proper correspondence between the textual description and visual material.
-
Conclusions Section: While the conclusions summarise the findings, they would benefit from the inclusion of more detailed and concrete recommendations for clinical practice. As it stands, the current conclusions do not sufficiently guide the practitioner in applying the findings effectively.
-
Suggested Text Removal (Lines 325–329): The sentence beginning with "In our study..." and continuing with a general statement about balance should be removed. This sentence does not offer meaningful conclusions and is more suitable for discussion rather than the conclusion section. Its presence in the final remarks weakens the practical utility of the conclusions.
In summary, the article is well-structured and contributes valuable knowledge to the field of physiotherapy and VR-based rehabilitation. I recommend its acceptance with minor revisions, specifically focusing on figure-text alignment, refinement of the conclusion section, and removal of the non-conclusive statement currently in lines 325–329.
Reviewer 2 Report
Comments and Suggestions for Authors
The article submitted for review may be important to readers of the Journal of Functional Morhology and Kinsiology. The title of this article reflects its content, although the study is more concerned with healthy individuals than clinical cases. The abstract correctly indicates the content of the work except for the introduction. In the opinion of the reviewer, the introduction does not sufficiently introduce the topic of the article, its research context and significance. In the introduction, the authors focused on clinical cases, and the study concerns healthy individuals. The authors should add a paragraph in which they should present how such combined therapy affects healthy individuals. In the last paragraph of the introduction, the authors stated the objective as "This research explores the potential and applicability of VR in clinical practice, with findings that may contribute to a better understanding of the influence of VR on postural control.", and only as the second objective did they state that "An additional motivation for conducting this study was the recurrent reports of instability and dizziness observed in previous clients following their first VR training session on the Icaros® system", while in the material/methods section they correctly describe the group of healthy people and there is no group of people with disorders presented there. In the opinion of the reviewer, after reading the entire work, this is justified. The authors focused their research on the impact of the method - which, as they themselves noted in the last sentence "This allowed for assessment of the immediate impact of the VR-based intervention on postural control" of part 2.2. The reviewer's attention was also drawn to figure 1 presented in parts 2.2.1 and 2.2.2. The reviewer would ask the authors to once again correctly assign figures 1 to 3 to the description in the text. The results of the experiment are presented clearly and legibly in tables and diagrams. The discussion relates the research results, including existing knowledge, and is conducted correctly. It should be noted that in the discussion the authors have already clearly justified the choice of the study group and the potential effects in the case of using the method not for training healthy people but for physical therapy for sick people, something that was missing in the introduction. However, in conclusion the authors again emphasize the impact of the therapy on sick people and not on the training aspect of healthy people. In the description of the research group, the authors did not present that these people had pain problems, a tendency to fall, etc. before taking part in the experiment. In the reviewer's opinion, the introduction should be expanded to include an element of using the tested method for training healthy people and not only focusing on the description of therapeutic use, but in the conclusions, focus only on the effects presented in the research work and possibly mention the potential effects in sick people based on literature.
Round 2
Reviewer 2 Report
Comments and Suggestions for Authors
After implementing the proposed changes, I have no further comments.